# GA4GH Passport standard for digital identity and access permissions

## Graphical abstract

## Highlights

- The GA4GH Passport is an open standard to encode data access permissions of users

- The standard enables communication of data access permissions at international scale

- Passport facilitates data access and analysis across international platforms

- Implementers include research initiatives, infrastructures, and national institutes

## Authors

Craig Voisin, Mikael Linden,
Stephanie O.M. Dyke, ..., Ilya Tulchinsky,
Jaime M. Guidry Auvil, Tommi H. Nyrönen

## Correspondence

craigv@google.com (C.V.),
tommi.nyronen@csc.fi (T.H.N.)

## In brief

Voisin et al. report the GA4GH Passport, a new international standard to encode machine-readable data access permissions for individual users. Passports are used as part of a federated data regulatory process to authenticate and authorize data users in managing access to human biomedical datasets and have been successfully implemented in international research programs and data infrastructures.

 Voisin et al., 2021, Cell Genomics 1, 100030
November 10, 2021 © 2021 The Author(s).

# Cell Genomics

## Technology

# GA4GH Passport standard for digital identity and access permissions

Craig Voisin,[1,15,16,*] Mikael Linden,[2,3,15] Stephanie O.M. Dyke,[4,15] Sarion R. Bowers,[5,15] Pinar Alper,[13]
Maxmillian P. Barkley,[9] David Bernick,[6] Jianpeng Chao,[1] Mélanie Courtot,[10] Francis Jeanson,[11] Melissa A. Konopko,[5,7]
Martin Kuba,[12] Jonathan Lawson,[6] Jaakko Leinonen,[2] Stephanie Li,[6,7] Vivian Ota Wang,[8] Anthony A. Philippakis,[6]
Kathy Reinold,[6] Gregory A. Rushton,[6] J. Dylan Spalding,[2,3] Juha Törnroos,[2,3] Ilya Tulchinsky,[14,15] Jaime M. Guidry Auvil,[8]
and Tommi H. Nyrönen[2,3,*]

[1]Google LLC, Kitchener, ON N2H 5G5, Canada
[2]CSC–IT Center for Science, Espoo 02101, Finland
[3]ELIXIR Finland, Espoo 02101, Finland
[4]McGill Centre for Integrative Neuroscience, McGill University, Montreal, QC H3A 2B4, Canada
[5]Wellcome Sanger Institute, Hinxton, Cambridgeshire CB10 1SA, UK
[6]Broad Institute of MIT and Harvard, Cambridge, MA 02142, USA
[7]Global Alliance for Genomics and Health, Toronto, ON M5G 0A3, Canada
[8]National Cancer Institute, National Institutes of Health, Bethesda, MD 20892, USA
[9]DNAstack, Toronto, ON M5H 1T1, Canada
[10]European Molecular Biology Laboratory, European Bioinformatics Institute (EMBL-EBI), Hinxton, Cambridgeshire CB10 1SD, UK
[11]Datadex Inc., Toronto, ON M5V 0C4, Canada
[12]Masaryk University, Brno 602 00, Czech Republic
[13]ELIXIR Luxembourg, Luxembourg Centre for Systems Biomedicine, University of Luxembourg, 4367 Belvaux, Luxembourg
[14]Verily Life Sciences, South San Francisco, CA 94080, USA
[15]These authors contributed equally
[16]Lead contact
*Correspondence: craigv@google.com (C.V.), tommi.nyronen@csc.fi (T.H.N.)

## SUMMARY

The Global Alliance for Genomics and Health (GA4GH) supports international standards that enable a federated data sharing model for the research community while respecting data security, ethical and regulatory frameworks, and data authorization and access processes for sensitive data. The GA4GH Passport standard (Passport) defines a machine-readable digital identity that conveys roles and data access permissions (called "visas") for individual users. Visas are issued by data stewards, including data access committees (DACs) working with public databases, the entities responsible for the quality, integrity, and access arrangements for the datasets in the management of human biomedical data. Passports streamline management of data access rights across data systems by using visas that present a data user's digital identity and permissions across organizations, tools, environments, and services. We describe real-world implementations of the GA4GH Passport standard in use cases from ELIXIR Europe, National Institutes of Health, and the Autism Sharing Initiative. These implementations demonstrate that the Passport standard has provided transparent mechanisms for establishing permissions and authorizing data access across platforms.

## INTRODUCTION

Genomic and health data research heavily relies on data reuse (also known as secondary use) and analysis beyond the original study's purpose. Secondary data analyses enable discoveries beyond the original studies and represent a paradigm shift from research that traditionally relied on investigators generating primary data for each research project. This requires data users to discover and potentially combine analyses across available datasets worldwide while adhering to the ethical, legal, and regulatory frameworks, expectations, and requirements that govern data access within organizations and across national and international boundaries.

The Global Alliance for Genomics and Health (GA4GH) has driven progress in sharing biomedical data.[1] However, barriers for data users and data stewards persist in our very connected global community.[2] A significant challenge to accelerating genomics-based research is the ability of organizations to manage data access processes and verify user identity and permissions. For data users, discovering and gathering sufficient data to address a hypothesis and then gaining data access approval can be time consuming. Processes and rules for accessing

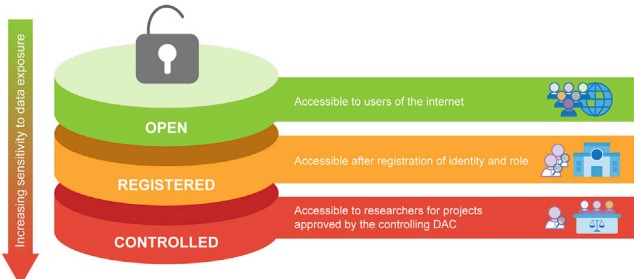

**Figure 1. Tiers of data access**
Datasets are commonly shared in databases with either open, registered, or controlled access, depending on the regulatory requirements. With higher regulatory requirements to control access, there is a need for increased security measures when accessing the dataset.

data vary, not only across repository boundaries and computing environments, but also by jurisdiction and organizations[3,4] (https://www.ncbi.nlm.nih.gov/projects/gap/cgi-bin/about.cgi; https://covid.cd2h.org/dur; https://gdc.cancer.gov/access-data/obtaining-access-controlled-data). Understanding and navigating these environments and regulations can be difficult for both data users and infrastructure providers. This is compounded by the often manual data access review and approval processes performed by data access committees (DACs).

The GA4GH Passport standard (https://github.com/ga4gh-duri/ga4gh-duri.github.io/blob/master/researcher_ids/ga4gh_passport_v1.md) defines a machine-readable digital identity, in the form of Passport visas, that communicates roles and data access rights of a data user granted by a DAC or other authority and enhances data user interoperability between data repositories.[5,6] These visas are used by technical infrastructure services to manage data access permissions of data users across repositories by conveying the permissions from their authority to the environment where data access takes place. A visa provides a technical representation of data governance process outcomes with a durable assurance that the person(s) accessing the data have been authorized to do so.

The GA4GH Passport standard is used in conjunction with other GA4GH standards. In particular, the GA4GH Authentication and Authorization Infrastructure (AAI) specification defines a standard for authenticating the identity of data users and delivering their Passports. While the Passport standard defines the syntax for expressing roles and data access rights, the AAI standard defines how they are exchanged between the parties.

Examples of databases for sensitive data sharing include the European Genome-Phenome Archive (EGA)[3] and the National Institutes of Health (NIH) Database of Genotypes and Phenotypes (dbGaP).[7] They are already supporting the GA4GH Passport standard in privacy-preserving data infrastructures that communicate access rights for datasets they hold.

The GA4GH Steering Committee approved the GA4GH Passport standard in 2019 (https://www.ga4gh.org/news/ga4gh-passports-and-the-authorization-and-authentication-infrastructure/). GA4GH technical standards are proposed and certified through the GA4GH product development and approval process.[1] The GA4GH Passport standard is available from Gi-

thub under the GA4GH Copyright Policy (https://www.ga4gh.org/wp-content/uploads/GA4GH-Copyright-Policy-Updated-Formatting.pdf) to any research organization or data provider. Organizations may adapt the specification into their own processes, as long as they maintain system interoperability as required by the standard.

## DESIGN

### Requirements

This section introduces the issues limiting genomic data sharing that prompted the design of the GA4GH Passport standard. Genomic data sharing has been limited by concerns about protecting patient privacy, security, and integrity of data. This is further complicated by the cost, logistics, and concerns about copying or moving data across organizational or international boundaries.

#### *Tiers of access*

The GA4GH Passport standard supports a three-tiered data-access model for data sharing:[8,9] open access, registered access, and controlled access to address a range of access permissions (Figure 1).

Open access (public access) refers to publicly available data. Most commonly, anyone can access these datasets anonymously or without specific permission—provided that they agree to open access licenses if such a license is in effect for the data. For example, Creative Commons defines the CC0 license (https://creativecommons.org/share-your-work/public-domain/cc0/), which enables data producers and owners "to waive those interests in their works and thereby place them as completely as possible in the public domain."

Registered access[10,11] models are a type of role-based access to datasets. Data providers commonly use a vetting process to determine if the individual is a "bona fide researcher" and a sign-in process to authenticate the user's identity. An "attestation" from the data user, linked to their institution, is submitted to document the data user's agreement to data use conditions. Once registered, these authenticated bona fide researchers can access and search data and services in the registered access tier. Such an access tier assists data users by allowing them to search more detailed information about datasets compared to anonymous internet users, improving their ability to find datasets relevant to their research. For example, registered access to data can be used in the data discovery phase or to give access to detailed summary-level data that cannot be given for anonymous users; i.e., it provides a technology and regulatory model to streamline access where data infrastructures managing the sensitive data and visa-issuing organizations are mutually recognized.

Controlled access models enable "oversight and investigator accountability for sensitive data sets involving personal health information."[3] In the implementation of controlled access, data stewards first make the curated data and discovery metadata available for authorized users; this is followed by three phases: data discovery, DAC review, and data use (Figure 2).

In the data discovery phase, the data user identifies the controlled access datasets of interest for their research. Once relevant data have been identified, the data user submits a

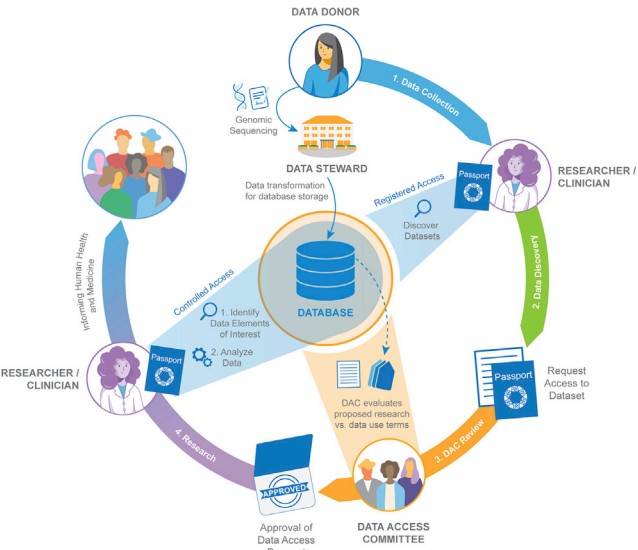

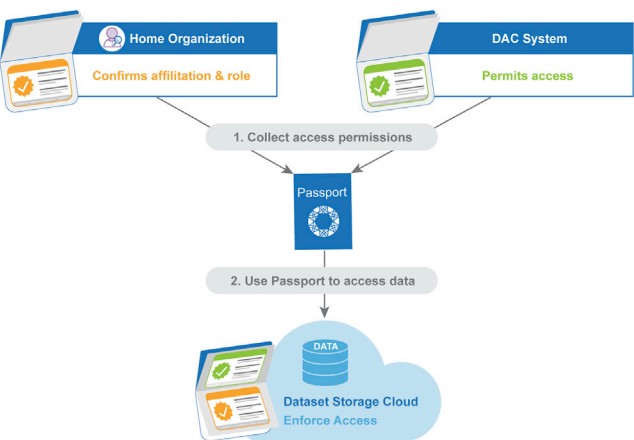

**Figure 2. Passports in action**

Passports provide interoperable infrastructure to authenticate and authorize users across biomedical services at various stages of a research project or study. Once data are deposited in a database they are then made available to other users. Users may be able to access part, or all, of the dataset through a registered access mechanism using their Passport and visa(s), where these provide the necessary credentials for access. For datasets managed with controlled access, the user will need to submit a DAR to the DAC. If and when the DAR is approved, the DAC will issue a visa granting access to the dataset(s). The data user will then apply for access to the database and provide the visas, attesting their identity and access permissions, which are verified in order to gain access to the data.

data access request (DAR) to the DAC for review. In the DAC review phase, the DAC must verify the identity of the data user and determine if the proposed research is within the bounds of the permitted use(s) of the dataset. If approved, the data user and their institution must agree to the terms of use of the repository's data through a data use or processing agreement. In the data use phase, the data user gains access to the dataset(s). If the datasets are locally stored, the data user downloads the data to their own computing environment for their data analyses. If federated data sharing is used, the user may analyze in a shared computing environment.

### Federated data sharing

The GA4GH Passport standard is one component used to support a paradigm change where computational data analysis can be transported to the platform where the data are accessible. This is essential for federated data sharing models, where datasets are hosted across multiple secure computing services or clouds (Figure 3, see also Thorogood et al. in this issue[9]). Whether a dataset can be made available for a user on an open, registered, or controlled access tier depends on the sensitivity of the data and regulatory requirements. The data stewards make the datasets available for data users with proper data access permissions granted by a competent DAC.[9] It is a responsibility of the environment to enforce data access, i.e., to make sure that only data users with proper data access permissions can access the data.

**Figure 3. Enabling federated data sharing**

A data user receives data access permissions from a DAC and can run their analysis in one or more computing environments (e.g., a cloud) which have a copy of the dataset. A Passport relays the data access permission granted by the DAC to the computing environment which enforces the data access rights, only granting access to the data users with proper data access permissions.

The environment where data are analyzed can reside on a public commercial cloud, if permitted by the data steward and applicable laws, but also on private clouds or dedicated high-performance computing environments.

*Data remains in defined locations.* When data are made accessible in a secure environment, the data no longer need to be downloaded outside of that environment to perform analysis. Copies allowed by the data stewards on such services give data users options for where they can analyze data, potentially even across datasets. Copies may be made available to authorized users in different secure cloud environments and across geographical locations within compliance constraints. In particular, if there is a regulatory requirement that the data must not leave a particular jurisdiction, the authorized users may analyze the data using services within that jurisdiction and export only the results.

*Compute jobs move to cloud.* Leveraging cloud computing environments means the data user may configure the data analysis jobs to run near the data center hosting the data. There are other emerging GA4GH standards, such as the Workflow Execution Service (WES),[1] that define how workflows can be made portable, moved to, and executed in an environment where the data are available for authorized data users.

### Integrity and authenticity of access rights

As described above, the data access rights are granted by a data steward, such as a DAC for controlled access data, and consumed in an environment where data access permissions are enforced. The GA4GH Passport standard must ensure the data access rights' integrity (that they were not tampered with) and authenticity (that they were granted by their authoritative source) while they traverse to the computing environment, potentially through several intermediary systems.

### Leverage existing industry standards

The GA4GH Passport standard relies on existing industry standards, as these standards have a range of established

**Table 1. GA4GH Passport visa types**

| Visa type | Common use case | Description |
|---|---|---|
| ControlledAccessGrants | Controlled access tier | Describes the data user's permission to access controlled access datasets that commonly results from an approval by a DAC |
| AcceptedTermsAndPolicies | Registered access tier (extensible to other use cases) | Describes the data user has acknowledged specified terms, policies, and conditions that are necessary for accessing data or a service. The terms, policies, and conditions are defined by a specific standard referenced in the visa. e.g., attestation for registered access |
| ResearcherStatus | Registered access tier (extensible to other use cases) | Describes the data user as a researcher or clinical care professional as defined by a specific standard referenced in the visa |
| AffiliationAndRole | Allows implementing access or restrictions based on affiliation and role | Describes the data user's role within their institution, e.g., faculty@cam.ac.uk indicates the person is a faculty member in the University of Cambridge |
| LinkedIdentities | Provides source of authority to unify multiple accounts for one individual | Describes the different user identifiers the data user is confirmed to have, e.g., "lisa.z@elixir-europe.org" and "lisaz@csc.fi" represent the same person |

The five Passport visa types in the GA4GH Passport Standard express the data user's digital identity, roles, and organizational affiliation, data access rights obtained from DACs, and linked identities for the same user across various systems or institutions. The "Common use case" and "Description" columns highlight how different visa types in the GA4GH Passport specification are commonly used.

implementations (both commercial and open source) available for the deployers to choose from. Leveraging established standards supports secure deployments, as these mature standards are generally considered to have a proven track record and users will be familiar with their use, thereby reducing flaws and surface area for attack.

### Design of GA4GH Passports

This section introduces the design of the GA4GH Passport standard, which is based on the requirements outlined in the previous section. The section first presents the Passport visas, which are atomic assertions about a data user's roles and permissions. Then the Passport system components and the accompanying AAI specification are introduced to describe how visas flow from their sources (e.g., a DAC) to the environment where the data access takes place (e.g., a cloud).

#### *Passport visas*

The GA4GH Passport standard provides a standardized mechanism for data users to present their digital identity, including authenticated credentials and permissions, in the form of visas and to share this across distributed data systems and organizational boundaries. Using this framework ensures a common language and improves adherence to privacy, security, and data access policies. Requests for open, registered, and controlled access data (see Tiers of access) are managed effectively.

Figure 2 provides an overview of how the GA4GH Passport standard (Passport) is used to represent the data user's digital identity. After discovering data of interest, the user authenticates their electronic identity and researcher status. For some data this may be sufficient for the data user to gain access to services in the registered access tier, but commonly the user must make a DAR to one or more DACs to obtain access to all the data. The GA4GH Passport collects the granted data access permissions as the process proceeds (Figure 2). The Passport communicates the permissions granted as visas to a service where data can be made available for analysis. An example of a Passport and how it is used is provided in the Supplemental information. All visas

have expiration dates and may be revoked based on changes in the data user's status (https://github.com/ga4gh-duri/ga4gh-duri.github.io/blob/master/researcher_ids/ga4gh_passport_v1.md#token-revocation): for example, when a data user changes institutions.[12]

Table 1 describes the five Passport visa types and how they relate user permissions to dataset access. To match the requirements presented in the previous section, the GA4GH Passport Standard defines five visa types to express the data user's digital identity, roles, organizational affiliations, data access rights obtained from DACs, and linked identities for the same user across various systems or institutions.

For controlled-access data resources, the data user needs permission to access the datasets. This visa commonly results from an approval by a DAC. The DAC approves a DAR and issues a ControlledAccessGrants visa to communicate that approval to the data management system.

For data held in a registered access tier, visas communicate the qualifications of a data user to obtain access to registered access data and services. The bona fide researcher status, defined in the Requirements section, is expressed using a ResearcherStatus visa. The attestations are expressed using an AcceptedTermsAndPolicies visa, and these convey that the data user has acknowledged specified terms, policies, and conditions that are necessary for accessing data or a service.

The AffiliationAndRole visa type can be used to describe the data user's organizational affiliation(s) and professional role. For example, a visa can be used to limit access to a service to members of a specified research collaboration. The absence of the visa can be used to exclude users who are no longer affiliated with the authorized collaboration or organization.

The GA4GH Passport standard also effectively manages multiple data-user identities (Figure 4). Data users are often affiliated with multiple institutions or projects and may have used different identities, as represented by identifiers such as usernames or e-mail addresses, for various tasks. For example, a data user may have an identity approved by a DAC that permits accessing

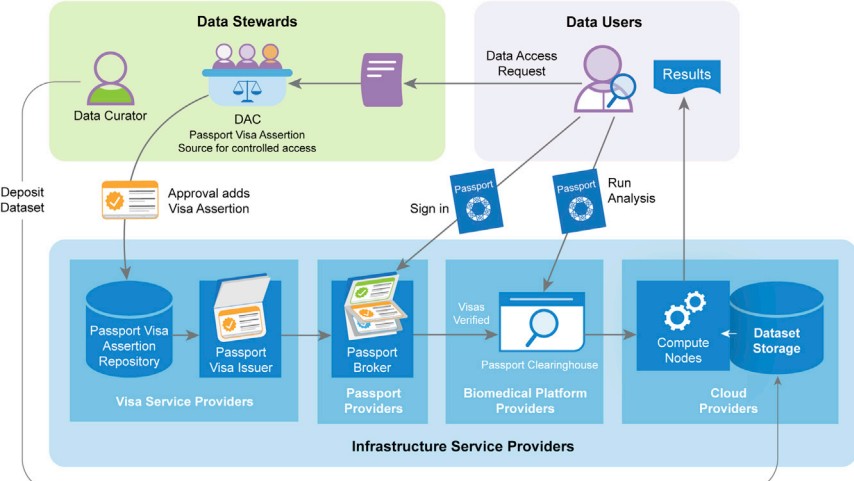

**Figure 4. Linking identities in Passports**
Home organizations and DACs identify their data users with identifiers (green elements in the lower part of the figure), which are minted in the visas they issue. Cloud environments issue their own user identifiers (green elements in the upper part of the figure). The LinkedIdentities visa type is used for signaling that particular identifiers belong to the same person (green lines in figure).

datasets from an archive, another identity for the computing environment where the analysis takes place, and a third one for the affiliated home organization. This is cumbersome for the data user and introduces barriers for automating analysis and managing access control.

To address this challenge, the GA4GH Passport standard supports identity linking. For instance, a data user receives from a DAC a ControlledAccessGrants visa that contains their user identity as it was presented to the DAC in the DAR. The data user also receives an AffiliationAndRole visa that carries their user identity from their home organization. To confirm these two visas belong to the same individual, a third visa, called LinkedIdentities, is issued to map the two user identifiers. To receive a LinkedIdentities visa, the user needs to demonstrate that they are the holder of both identities.

All visas are represented as digitally signed tokens (a JSON Web Token, see Supplemental information section). The digital signature is created by the Passport Visa Issuer on behalf of the Passport Visa Assertion Source, as described in the next section.

### Passport system components

This section describes the system components that the GA4GH Passport standard introduces, including visas, brokers, and clearinghouses. The system components can be integrated into existing systems and processes to make organizations technically interoperable with the GA4GH Passport. The basic flow of visas, originating from data stewards who issue the visas and continuing to Passport Clearinghouses who receive and evaluate visas, is depicted in Figure 5 and explained below.

A Passport Visa Assertion Source is an organization that acts as the source of authority to assert, authorize, or attest to terms of use required for data access. For the ControlledAccessGrants visa as described above, the Passport Visa Assertion Source is the DAC, whereas for ResearcherStatus and AffiliationAndRole visas it is typically the data user's home organization. Assertions are then recorded in a Passport Visa Assertion Repository.

A Passport Visa Assertion Repository stores assertions and provides an interface to record and query them. For a Controlle-

dAccessGrants visa, it is typically the data archive's database where the DAC records the granted data access permissions. Database technology and format are implementation-specific.

A Passport Visa Issuer queries the Passport Visa Assertion Repository, formats, and digitally signs assertions in a verifiable GA4GH Passport visa format and makes them available to Passport Brokers. The visa contents and format were described in the previous section (see Supplemental information for an example).

A Passport Broker bridges the Visa Issuers and Passport Clearinghouses. A Passport Clearinghouse needs to know who the data user is and what visas they possess, and consequently the Passport Broker resolves this by authenticating the user, potentially using the federated identity management services provided by their home organization[5,6] (https://github.com/ga4gh/data-security/blob/master/AAI/AAIConnectProfile.md). After authenticating the user successfully, the Passport Broker pulls their visas from one or more Passport Visa Issuers and packages them to be delivered to the Passport Clearinghouse.

The exact interaction between the Passport Broker and the Passport Clearinghouse is defined in another GA4GH standard, called the GA4GH AAI Specification. The GA4GH AAI specification relies on OpenID Connect (https://openid.net/specs/openid-connect-core-1_0.html), a widely used federated identity management standard protocol, which builds on the OAuth2 standard (https://datatracker.ietf.org/doc/html/rfc6749) for authorizing access to application programming interfaces (APIs), such as those defined by other GA4GH standards. Therefore, OpenID Connect was found to fit well with the federated approach to data sharing (https://gdc.cancer.gov/access-data/obtaining-access-controlled-data).

A Passport Clearinghouse receives and evaluates Passport content for the access policies of the organization and decides to approve or deny access to the dataset copy it has. Passport Clearinghouses provide the flexibility to encode various policies—sometimes requiring several different types of visas—to meet specific access requirements. The digitally signed visas available via the Passport can be checked to see if they arrive unaltered, thereby retaining the original authority of the issuer. If any of the policy requirements are not met, or the authenticity of the Passport content cannot be verified, access is denied by default.

### Passports with visas from multiple issuers

A GA4GH Passport provides a collection of one or more visas of the same or different visa types, potentially issued by several

**Figure 5. GA4GH Passport system components**

In a controlled access scenario, the data user submits a DAR (purple document) to the DAC for approval. When the DAC approves the request, a visa assertion is added to the Passport Visa Assertion Repository (lower left). When the data user signs in to their account via the Passport Broker, the Visa Issuer loads their visa entries and signs them. The Passport broker collects the visas from the Visa Issuers, assembles the Passport, and gives it to the data user. When the data user accesses a Passport Clearinghouse, the Passport is included with requests such that the computing environment is made aware that access policies are met and access to the dataset can be permitted.

distinct Passport Visa Issuers. Enforcing data access based on the GA4GH Passports within a cloud computing environment, as enabled by the GA4GH Passport and AAI standards, offers flexibility because users are able to collect their visas from multiple sources of authority.

For example, a data user may have received a ControlledAccessGrants visa from a DAC. Typically, the data user's home institution and the organization hosting the DAC have signed a data access agreement that, among other things, has a condition that the data user's data access permission is revoked if they depart from their home institution. The data user selects the cloud where the datasets are available, signs in, and presents their Passport to access the data. The Passport may now contain two visas; the ControlledAccessGrants visa describes their permission to access the dataset, and the AffiliationAndRole visa describes their continuing affiliation with their home institution. The data access policy may require that both visas are present to justify the data access (Figure 6). The Passport bridges these two sources of authority to allow data access as intended by authorizing parties.

### Using Passports to enforce access

Organizations and systems can electronically verify data access permissions and authenticate data users across many datasets using the visas in the user's Passport. This provides the organizations and systems the ability to scale—by processing more access requests in a faster time frame—and allows for more automation of the process.

Many other GA4GH standards define an API to let clients programmatically access sensitive data, and these standards are used for discovery or streaming of data or execution of a workflow on it.[4] A Passport can be presented in the API call to authorize the access. In that scenario, the API is the Passport Clearinghouse that receives the Passport, validates it, and enforces the data access as described in the Passport system components section above.

Passports can also improve access control enforcement capabilities of existing software implementations leveraging Passport Broker services. For example, users of Galaxy[13] workflows can access datasets on a cloud[14] that uses the same protocols as in the Passport standard for identity and access manage-

ment. In that case, the Galaxy authenticates a user against a Passport Broker, receives their Passport, and presents it in a downstream API call to download the sensitive dataset for analysis. In this setup, the Passport Clearinghouse is the API that validates the Galaxy user's permission (typically, ControlledAccessGrants visa) to access the dataset before starting the streaming of the dataset to Galaxy.

### RESULTS

Since its approval as a GA4GH standard in October 2019, the GA4GH Passport standard has been adopted in three ecosystems: ELIXIR infrastructure in Europe, NIH in the United States, and Autism Speaks. GA4GH driver projects in Australia, Africa, Canada, and Japan are also under consideration for implementation. The GA4GH Passport provides access to data across repositories and biomedical platforms powered by ELIXIR, such as ELIXIR Beacon Network (https://beacon-network.elixir-europe.org), with data provided by the EGA. ELIXIR currently has 32 production or test data infrastructure services that consume data access information transmitted to them using Passports. Currently, the EGA manages and supports delivering visas asserted by 1,349 DACs.

### Implementations: Organizations, challenges, and benefits

Current implementations have shown similar challenges and benefits and provide examples for how other organizations can implement the GA4GH Passport to meet the challenges faced in federated data sharing (Table 2).

A research organization may play one or more of the following roles defined in Table 2: it can develop or manage research infrastructure (first role column), it can be the data steward for datasets (second role column), and/or it can carry out research projects (third role column). For example, the National Cancer Institute (NCI) in the U.S.:

Acts as an infrastructure service provider for storage and computing resources either directly or via contracts with third parties.

Acts as a data steward with policy and a DAC to manage access requests for many cancer-related datasets.

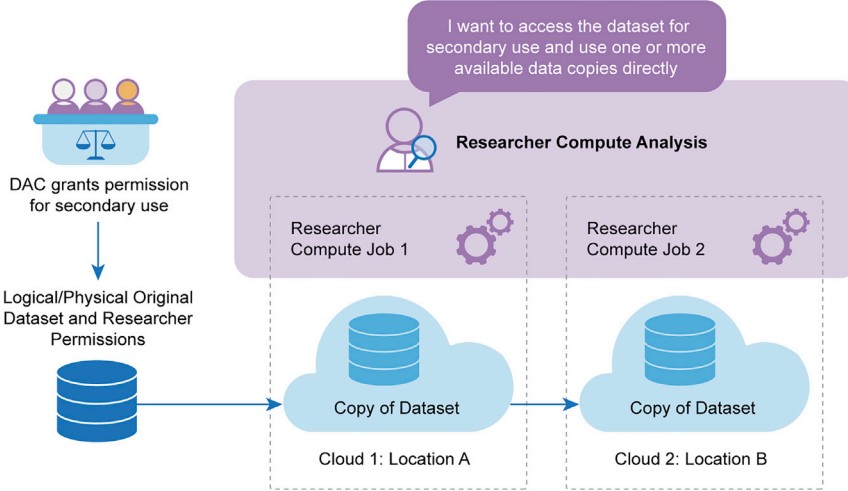

**Figure 6. Passports may contain visas from multiple sources**

Visas are used to describe data users' roles and access permissions. Users can collect multiple visas from various sources, including their institutional organization and data stewards via their corresponding DACs. These visas are collected into a data user's Passport, which is checked by the environment where the data access takes place to authenticate the user's identity and verify the access credentials needed to gain access to datasets.

Provides institutional support for data requests made by its local research data users.

## ELIXIR

ELIXIR offers strategic European life-science research data infrastructure of global significance. ELIXIR has brought Europe's national centers and core bioinformatics resources into a single, coordinated infrastructure since its launch in December 2013. ELIXIR AAI[15] is an identity and access management service portfolio used for authenticating data users and managing their access rights in services. ELIXIR

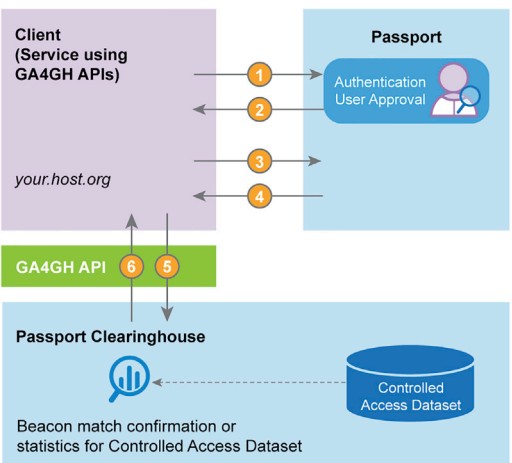

**Figure 7. Passport implementation protocols**

A technical protocol flow describing messages that are passed between Passport system components when a data user client accesses GA4GH Beacon services (ELIXIR Beacon Network, https://beacon-network.elixir-europe.org/). Figure 7 adds technical detail about how Passport implementation works to Figure 5 (Passport system components). Steps 1–4 (shown as arrows in Figure 7): "Authentication and authorization" as part of data user sign-in; step 5: "Run Analysis" or otherwise issue a request needing Passport authorization; step 6: "Results" returned as a response to the client.

AAI provides an implementation of a GA4GH Passport Broker (see Passport system components) that authenticates a data user with their ELIXIR identity and assembles their visas from various assertion repositories for delivery to downstream Passport Clearinghouses for the purpose of enforcing access to services.

The ELIXIR AAI Passport Broker can retrieve ControlledAccessGrants visas (see Passport visas) from the EGA,[3] a data infrastructure service for human genome-phenome datasets that have been created from samples collected under a consent that defines allowed data use, e.g. research. The EGA uses Passport to orchestrate data user access authorization to use those data and ensures compliance with the original consent. As a Visa Assertion Repository, the EGA manages its own database for data access permissions as granted by DACs. The DACs are able to upload their data access permissions to the EGA using the specified Passport visa format. For each controlled access dataset, there is a requirement for a specific visa issued from a specific DAC that is set as the requirement for granting access to that dataset. The EGA then provides the infrastructure to sign visas when data users request their ELIXIR Passport, and these visas are validated before access to data commences.[16]

The ELIXIR AAI Passport Broker is further integrated with the eduGAIN inter-federation (https://www.edugain.org), a research and education network service for cross-national federated login. eduGAIN enables the ELIXIR Passport Broker to authenticate users at their home institution, such as their home university, and retrieve their affiliation, expressed as an AffiliationAndRole visa. To support registered access, the ELIXIR Passport Broker and its back-end processes also manage ResearcherStatus and AcceptedTermsAndPolicies visas for ELIXIR identity holders.

## NIH

NIH is part of the U.S. Department of Health and Human Services. Many of the institutes and centers within NIH store genomics and health data and manage data access approvals. The dbGaP was developed to archive and distribute the data and results from studies and, as a result, stores DAC decisions regarding data access. The dbGaP enables many use cases within NIH for intramural (or internal) research use, as well as

**Table 2. The benefits of implementing the GA4GH Passport**

| | Infrastructure service providers | Data steward | Data users |
|---|---|---|---|
| Role definition | Provides common data access support mechanisms on cloud infrastructure, including high-performance computing, and may model policy or regulatory frameworks via software services. | Organizes research initiatives, provides oversight, and authorizes data access for secondary use as a data steward (sometimes using infrastructure of approved 3rd party organizations). | Proposes studies and conducts research making use of shared data available from data stewards once their studies are approved for such use. |
| Challenge | Connects multiple layers of disparate systems that use different identity and permission models that may not be compatible. It can be challenging to carry the data access authorization from a DAC through these layers to apply to the data being stored. | A need to convey authorization across disparate infrastructure systems, sometimes hosted by external organizations, while maintaining data governance oversight of the data user's use. Also may wish for a more secure and streamlined verification process during DAR reviews to ensure data users meet the consented restrictions agreed upon by the data donors using a policy and ethics framework. | Navigate policies and procedures to apply for data access and use data for custom data analysis with minimum hurdles while being able to leverage a variety of research infrastructure services available (e.g., data repositories, cloud computing, and home institution resources). |
| Passport benefit | Provides a mechanism that can securely collect and carry permissions to where they need to be checked at the data repository as part of software control layers while retaining digital proof of authority across systems and organizations. | Encodes access authorization to infrastructure service providers to verify that permission to access the data within a secure environment. Data users use the data in-place without the need to download and secure it separately. Infrastructure creates audit logs and other security features needed by data governance oversight. May carry digital signatures from data user organizations to the DAC to expedite the data use application. | Ability to communicate authority granted to a data user across organizational boundaries. The Passport carries this authority to data systems that provide secure access for users, and the same data access rights are implemented across computing environments. This removes the need to have copies of the data in local systems for processing, improving both data security and governance over the data. |

The use of the GA4GH Passport provides benefits across the stages of data sharing from providers to data users. A secure healthcare data sharing environment for research will include infrastructure service providers, data stewards, and DACs. Data users are researchers who use these services to access the shared datasets. The GA4GH Passport standard implementation for managing access to data will address challenges and provide benefits for each of these stages of the managed data access process.

extramural (i.e., including external collaborators) research approvals. These processes lay the groundwork for Passport support across many foundational projects and platforms in NIH and its collaborating organizations (https://www.ncbi.nlm.nih.gov/projects/gap/summaries/cgi-bin/darGrowth.cgi). To facilitate interoperability of technologies with these processes, NIH has introduced a Passport Broker called Researcher Auth Service (RAS). RAS is used across a broad spectrum of initiatives as the standard authentication and authorization mechanism for research going forward (https://datascience.nih.gov/researcher-auth-service-initiative). RAS provides common authentication and uses an authorization mechanism that is shared across institutes with a unified auditing capability while providing interoperability across NIH institutes. dbGaP is the first Passport Visa Assertion Repository integrated with RAS (https://auth.nih.gov/docs/RAS/serviceofferings.html).

Multiple biomedical platforms are in the process of implementing additional layers of Passport Brokers and Passport Clearinghouses that receive visas from dbGaP via RAS as part of their authorization checks. These platforms include Gen3 (https://gen3.org/; a data platform for building data commons and data

ecosystems from the University of Chicago), Sequence Read Archive (https://www.ncbi.nlm.nih.gov/sra; NIH's primary archive of high-throughput sequencing data), Terra (https://terra.bio; a scalable platform for biomedical research from the Broad Institute of MIT and Harvard), and SevenBridges (https://www.sevenbridges.com/promoting-interoperability-and-standardization-seven-bridges-and-ga4gh/; specializing in software and data analytics to drive public and private healthcare research). When this work completes, multiple clouds will be supported, including non-commercial cloud storage hosted by the National Library of Medicine (NLM; https://www.nlm.nih.gov) as well as commercial clouds provided by Google (https://cloud.google.com/gcp) and Amazon (https://aws.amazon.com).

### Autism Sharing Initiative (ASI)

ASI[17] aims to build the first federated and fully protected global network for sharing genomics and biomedical data to accelerate research toward the development of precision healthcare approaches for individuals with autism.

This global network is implementing the GA4GH Data Connect (https://github.com/ga4gh-discovery/data-connect/blob/

**Table 3. The GA4GH Passport implementations.**

| | Infrastructure service providers | Data stewards | Data users |
|---|---|---|---|
| ELIXIR | ELIXIR AAI provides a Passport Broker and issues visas based on its internal sources and via its integration with the data user's home organization's login systems. Also collects visas from EGA and DAC tools, such as the Resource Entitlement Management System (REMS) (https://github.com/CSCfi/rems). | As an ELIXIR Core data resource, EGA provides policy oversight functionality, such as access controls and audit logs, to over 1,000 DACs and data stewards that submit their access approvals to EGA (https://ega-archive.org/dacs). | ELIXIR AAI Passports support both intramural data users within the data steward organization and extramural data users that apply for data access. |
| NIH | Common Passport infrastructure is provided by NIH via dbGaP and RAS, while biomedical platforms that provide data hosting are provided by either specific NIH institutes or external partners (including SevenBridges, University of Chicago, The Broad Institute). | Each institute within NIH usually provides its own data governance and oversight policies for datasets for which it acts as the data steward, including a DAC that reviews and approves DARs. | Intramural data users (employed per institute) and extramural data users (from external organizations) may apply for access based on qualifying for secondary use of the dataset. |
| ASI | Each participating organization provides their own infrastructure, but they are used in combination by data users in a distributed setting. Some organizations host their data in public clouds while others host sin self-managed data centers. | Each participating organization creates and enforces its own policies on access to data (typically using a DAC). A policy framework is recommended by the consortium for safe data sharing policies between organizations. Inter-organizational policies, when agreed to by the participating organizations, are based on data users being approved by other member DACs. | Data users may have access to datasets from additional ASI organizations based on DAC approval from participating ASI organizations. Which datasets a data user has access to with a particular DAC approval is dependent on the policies in place by all data stewards involved. |

The GA4GH Passport specification has been implemented in three ecosystems: ELIXIR infrastructure in Europe, NIH in the United States, and Autism Speaks, supported by local and Google platforms. This table shows examples of data access information that is collected in these ecosystems and how they leverage the standard.

develop/SPEC.md) and Data Repository Service APIs (https://ga4gh.github.io/data-repository-service-schemas/preview/release/drs-1.2.0/docs/), with access control managed by the implementation of GA4GH Passport visas, to enable search and data analysis to be performed across multiple federated datasets with data remaining in the respective local environments. Members of ASI who manage access to the autism datasets participate in the network by using GA4GH Passports in two important ways: (1) issuing Passport visas for research users who have been approved through the dataset's DAC, and (2) making access decisions to a dataset based on Passport visas issued by other data stewards.

ASI members maintain autonomy over their data by controlling the policies used to enforce access control on their data. Members can choose to grant access to research users who have gone through DAC processes of other members, or they can restrict access to less sensitive data depending on their institutional and legal obligations (https://www.digitalsupercluster.ca/programs/precision-health/autism-sharing-initiative/).

Autism Speaks, a member organization of ASI, has been using the GA4GH Passport as a mechanism for access control to their MSSNG dataset[18] since 2019. The MSSNG dataset contains thousands of sequenced genomes from families affected by autism. Initially, the Passport was used for authorization between

components serving MSSNG data; authenticated users are issued a Passport with controlled access visas to the MSSNG dataset, and other components validate these visas before granting access to data.[19]

As the ASI network develops, this technology used by Autism Speaks will be extended to support access control decisions based on visas issued by other institutions. Other institutions will adopt this technology to issue visas for their data users, and agreements between members will be enforced by access control policies that rely on visas as a secure evidence of DAC approval from other ASI members (Table 3).

## DISCUSSION

The GA4GH Passport standard enables data stewards to centralize their controlled data access decisions. The Passport standard can then be used to mediate the data access permission for access control enforcement to any computing environment where a copy of the dataset resides. Thus, the GA4GH Passport supports a paradigm shift toward the federated approach to data sharing, which enables sensitive data to stay in a secure environment chosen by the data steward. This is an improvement over the current practice in which a data user downloads the data to their own local computing environment,

where it is outside the data steward's technical controls and where the data steward can control it potentially only contractually. This is an important contribution for research on data derived from human subjects.

Many other GA4GH standards describe APIs for programmatic access to sensitive data and services, such as Beacon API for data discovery, htsget API for data streaming, and WES[1] for remote execution of a workflow.[12] The GA4GH Passport is an integral standard that many other GA4GH standards need to authorize API access. Work is ongoing in GA4GH to attain this goal of facilitating access to all GA4GH APIs that protect registered or controlled access data.

The GA4GH Passport standard was developed as a GA4GH product,[1] initially focused on human biomedical data sharing. The Passport also has potential to be used for sensitive data access management for other data types, such as digital imaging data. There are also similar three-tier data access models outside the biological and medical sciences, such as in linguistics (https://www.clarin.eu/content/licenses-and-clarin-categories). A wider application of the Passport standard might require donating it to a cross-discipline standards organization for further development and approval.

### Limitations of the study
The GA4GH Passport standard is a technical specification defining an interoperable presentation for roles and data access permissions of a data user, as described in the Design of GA4GH Passports section. The GA4GH AAI specification defines how users are authenticated and how their Passports presented to a Passport Clearinghouse, as described in the Passport system components section. However, these two standards on their own are not sufficient for defining a trust framework that helps the Passport Clearinghouse to decide which Passport Brokers or Visa Issuers can be trusted. In order to develop such trust frameworks, additional organizational, technical, and operational requirements need to be introduced for the organizations managing Passport Brokers and Visa Issuers and their assessment and certification. Though some of these requirements can be derived from the GA4GH Data Security Infrastructure Policy (https://github.com/ga4gh/data-security/blob/master/DSIP/DSIP_v4.0.md), the detailed requirements need to be agreed between the parties.

### Opportunities for further advances
Potential additional implementers of the GA4GH Passport standard in the near term include Canadian Distributed Genomics (CanDIG),[20] GEnome Medical alliance (GEM) Japan (https://www.amed.go.jp/en/aboutus/collaboration/ga4gh_gem_japan.html), Human Heredity & Health in Africa (H3Africa; https://h3africa.org/), and the EU 1+ Million Genomes Initiative (https://digital-strategy.ec.europa.eu/en/policies/1-million-genomes). An ecosystem of Passports implemented across international programs offers better data access compatibility between data environments. Combining access-controlled datasets to form larger datasets to analyze increases the statistical power of the study as well as the inclusion of a diversity of datasets. This becomes a powerful tool to allow researchers to create datasets with the necessary statistical power and diversity to support the

research needed for genomics to become a routine part of healthcare decision-making.

As genomic and biomedical data continue to accumulate and their research use grows, the burden of reviewing and approving DARs is increasing[3] (https://covid.cd2h.org/dur; https://gdc.cancer.gov/access-data/obtaining-access-controlled-data). Below are described some opportunities to further develop the GA4GH Passport standard introduced in this article to streamline the DAC review process, reducing the burden for both the data user and DAC members.

The Data Use and Researcher Identity work stream of GA4GH will be investigating the possibility that a DAC could make use of visas to recognize the data access permissions other DACs have already granted to a project. By having a degree of mutual recognition of each other's authorizations, a DAC could choose to apply a more lightweight review for a DAR that another DAC has approved for datasets with similar DUO codes.[21,22] The intended outcome is to streamline the workload of the DACs and reduce the barriers researchers face when attempting to access multiple datasets under the control of two or more DACs by effectively enabling one DAC to provide the basis of an approval for datasets under the control of another DAC.

Another area where the GA4GH Passport standard may be able to improve the process of DAR approval is with the signing process of data access agreements. Many DACs have a process where an institutional representative (sometimes called a "signing official") must (co-)sign a DAR. This may introduce a substantial lead time. Therefore, the GA4GH Passports standard team is considering the creation of a new type of visa, whereby the "signing official" could assert that a given Principal Investigator has an institutional approval to sign the data access agreement on behalf of that institution.[8]

### Conclusions
The GA4GH Passport standard enables DACs and the data stewards who issue visas to communicate data users' roles and data access permissions to the organization and environment where the data access takes place. Passports and visas provide the digital identity and permissions for data users to obtain access to data. They are designed to be flexible enough to meet a variety of regulatory policy needs, giving data stewards the control they need to define restrictions. The implementation of Passports to collect and communicate data access permissions on global human biomedical data infrastructure will allow researchers to access and jointly analyze datasets across organizations, computing environments, and jurisdictions.

### STAR★METHODS

Detailed methods are provided in the online version of this paper and include the following:

- KEY RESOURCES TABLE
- RESOURCE AVAILABILITY
  - Lead contact
  - Materials availability
  - Data and code availability
- METHOD DETAILS

## SUPPLEMENTAL INFORMATION

## ACKNOWLEDGMENTS

The authors would like to acknowledge the GA4GH DURI work stream. We are also grateful for contributions from the ELIXIR federated human data community to drive implementation of Passport services (Thomas Keane [EMBL-EBI], Venkata Satagopam [LU], Ilkka Lappalainen [FI], Jordi Rambla [ES], Gary Saunders and Serena Scollen [ELIXIR], Mallory Freeberg, Giselle Kerry, Jorge Izquierdo Ciges, and Ashutosh Shimpi [EMBL-EBI]) and the developers of Google/DNAstack Passport services (Marc Fiume, Milan Panik, Monica Valluri, Patrick Magee, and Viliam Ročkai) for the user interfaces and service integration related to the original two Passport implementations. We are grateful to Angela Page, Alice Mann, and Peter Goodhand (GA4GH), Ewan Birney (EMBL, GA4GH), Ian Fore (NCI), Brian O'Connor (Broad Institute), Jeremy Adams (GA4GH), Michele Mattioni (Seven Bridges), Jiaqi Liu (University of Chicago), and Jamal Nasir (Univ. Northampton) for their support.

The authors would like to extend a special thanks to Kurt Rodarmer (NCBI/NLM) for all his technical contributions to the Passports specification and its use cases as well as consultations across GA4GH, including his support as part of the preparation of this paper. T.H.N., J. Leinonen, and J.T. were funded by the Academy of Finland grant no. 319968 and the ELIXIR Europe 2019-2023 program, which also funded M.K. C.V., J.C., and I.T. were funded by Google LLC. M.L. was funded by the CINECA project (H2020 No. 825775). S.O.M.D. was funded by CanSHARE and the Canadian Open Neuroscience Platform (CONP). S.R.B. was funded by Wellcome Trust grant number 206194. J. Lawson, K.R., D.B., and G.A.R. were funded by the Broad Institute of MIT and Harvard. M.P.B. was funded by DNAstack. M.C. and J.D.S. were funded by EMBL-EBI Core Funds, Wellcome Trust GA4GH award number 201535/Z/16/Z, and the CINECA project (H2020 No. 825775). F.J. was funded by University Health Network. S.L. and M.A.K. were funded by GA4GH. A.A.P. was funded by the Broad Institute of MIT and Harvard and the AnVIL Data Ecosystem (NHGRI-U24HG010262). P.A. was funded by ELIXIR Luxembourg.

## AUTHOR CONTRIBUTIONS

All authors contributed to investigation and writing – review and editing. J.M.G.A., T.H.N., C.V., M.L., S.O.M.D., D.B., J. Lawson, and I.T. contributed to conceptualization. C.V., S.O.M.D., S.R.B., M.P.B., D.B., J.C., J. Leinonen, J.T., I.T., and M.K. contributed to validation. C.V., M.L., S.O.M.D., S.R.B., J. Lawson, D.B., M.K., J.D.S., and I.T. contributed to methodology. C.V., M.P.B., J.C., M.K., J.D.S., J.T., and I.T. contributed to software. S.L., C.V., S.O.M.D., S.R.B., and J. Lawson contributed to visualization. C.V., M.L., S.O.M.D., S.R.B., M.A.K., and I.T. contributed to project administration. C.V., S.O.M.D., S.R.B., J.M.G.A., T.H.N., M.A.K., and I.T. contributed to supervision. J.M.G.A., T.H.N., C.V., M.L., S.O.M.D., and S.R.B. contributed to writing – original draft.

## DECLARATION OF INTERESTS

A.A.P. is a venture partner at GV and an employee of Alphabet Corporation and has received funding from MSFT, Verily, IBM, Intel, Bayer, and Novartis.

## WEB RESOURCES

Database of Genotype and Phenotype (dbGaP) Data Submission Policies, https://www.ncbi.nlm.nih.gov/projects/gap/cgi-bin/about.cgi
Data Use Request (DUR) Process, https://covid.cd2h.org/dur.

Obtaining Access to Controlled Data, https://gdc.cancer.gov/access-data/obtaining-access-controlled-data
GA4GH Passports and the Authorization and Authentication Infrastructure, https://www.ga4gh.org/news/ga4gh-passports-and-the-authorization-and-authentication-infrastructure/.
Creative Commons Public Domain (CC0), https://creativecommons.org/share-your-work/public-domain/cc0/
Token Revocation, https://github.com/ga4gh-duri/ga4gh-duri.github.io/blob/master/researcher_ids/ga4gh_passport_v1.md#token-revocation
eduGAIN – enabling worldwide access, https://www.edugain.org
GA4GH Authentication and Authorization Infrastructure (AAI) OpenID Connect Profile (DRAFT RFC), https://github.com/ga4gh/data-security/blob/master/AAI/AAIConnectProfile.md
OpenID Foundation, OpenID Connect Core 1.0 incorporating errata set 1, https://openid.net/specs/openid-connect-core-1_0.html
Internet Engineering Task Force, The OAuth 2.0 Authorization Framework, https://datatracker.ietf.org/doc/html/rfc6749
ELIXIR Beacon Network, https://beacon-network.elixir-europe.org/
Summary Statistics of dbGaP Data, https://www.ncbi.nlm.nih.gov/projects/gap/summaries-cgi-bin/darGrowth.cgi
Researcher Auth Service Initiative | Data Science at NIH, https://datascience.nih.gov/researcher-auth-service-initiative
Research Auth Service (RAS)- Service Offerings, https://auth.nih.gov/docs/RAS/serviceofferings.html
Welcome to Gen3, https://gen3.org/
Home - SRA – NCBI, https://www.ncbi.nlm.nih.gov/sra
Terra.Bio: Home, https://terra.bio/
Promoting Interoperability and Standardization: Seven Bridges and GA4GH - Seven Bridges, https://www.sevenbridges.com/promoting-interoperability-and-standardization-seven-bridges-and-ga4gh/
National Library of Medicine - National Institutes of Health, https://www.nlm.nih.gov
Cloud Computing, Hosting Services, and APIs, https://cloud.google.com/gcp
Amazon Web Services (AWS) - Cloud Computing Services, https://aws.amazon.com
GA4GH Data Connect API, https://github.com/ga4gh-discovery/data-connect/blob/develop/SPEC.md
GA4GH Data Repository Service API [Internet], https://ga4gh.github.io/data-repository-service-schemas/preview/release/drs-1.2.0/docs/
Autism Sharing Initiative, https://www.digitalsupercluster.ca/programs/precision-health/autism-sharing-initiative/
EGA, Browse DACs, https://ega-archive.org/dacs
GA4GH Passport, https://github.com/ga4gh-duri/ga4gh-duri.github.io/blob/master/researcher_ids/ga4gh_passport_v1.md
CLARIN – the research infrastructure for language as social and cultural data. Licenses and CLARIN categories, https://www.clarin.eu/content/licenses-and-clarin-categories
Global Alliance for Genomics and Health. Data Security Infrastructure Policy. Version 4.0, https://github.com/ga4gh/data-security/blob/master/DSIP/DSIP_v4.0.md
GEM Japan (GEnome Medical alliance Japan) | Japan Agency for Medical Research and Development, https://www.amed.go.jp/en/aboutus/collaboration/ga4gh_gem_japan.html
H3Africa | Human Heredity & Health in Africa | Human Genomic Research, https://h3africa.org/
European "1+ Million Genomes" Initiative, https://digital-strategy.ec.europa.eu/en/policies/1-million-genomes

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

## STAR★METHODS

### KEY RESOURCES TABLE

| RESOURCE | SOURCE | IDENTIFIER |
|---|---|---|
| **Software and algorithms** | | |
| ELIXIR AAI GA4GH echo service source code | ELIXIR, Linden et al. F1000 Research 2018. | https://github.com/ICS-MU/ga4gh-passport-tester |
| MitreID ProxyIdP OIDC Server source code | MitreID | https://doi.org/10.5281/zenodo.4877248 |
| CESNET Perun Project source code | CESNET | https://doi.org/10.5281/zenodo.5032084 |
| GA4GH Claim Source code | ELIXIR, Linden et al. F1000Research 2018. | https://github.com/CESNET/perun-mitreid/blob/master/oidc-idp/src/main/java/cz/muni/ics/oidc/server/elixir/GA4GHClaimSource.java |
| GA4GH Token Parser code | ELIXIR, Linden et al. F1000Research 2018. | https://github.com/CESNET/perun-mitreid/blob/master/oidc-idp/src/main/java/cz/muni/ics/oidc/server/elixir/GA4GHTokenParser.java |
| ELIXIR REMS source code | ELIXIR Finland (CSC), Linden et al. F1000Research 2018. | https://github.com/CSCfi/rems |
| **Other** | | |
| ELIXIR registration service | ELIXIR, Linden et al. F1000Research 2018. | https://elixir-europe.org/register |
| ELIXIR profile page | ELIXIR, Linden et al. F1000Research 2018. | https://profile.aai.elixir-czech.org |
| ELIXIR sensitive data access management test system | ELIXIR Finland (CSC), Linden et al. F1000Research 2018. | https://sd-apply.csc.fi/ (beta) |
| ELIXIR echo test service | ELIXIR, Linden et a al. F1000Research 2018. | https://echo.aai.elixir-czech.org/ |
| GA4GH Passport Standard v1.0 | GA4GH, *this manuscript* | https://bit.ly/ga4gh-passport-v1 |
| GA4GH AAI Profile Standard v1.0 | GA4GH | https://bit.ly/ga4gh-aai-profile |
| GA4GH Passport Visa Format v1.0 | GA4GH, this manuscript. | https://bit.ly/ga4gh-visa-format-v1 |
| GA4GH Passport Overview Video | GA4GH | https://www.google.com/url?q=https://youtu.be/selSbfNmOyw&sa=D&source=docs&ust=1634328343475000&usg=AOvVaw085tDPDWFfJPw7FCluHb_L |
| Webinar GA4GH Passports: Benefits of Integrating a Global Electronic ID for Accessing Biomedical Data (Part 1) Summary | GA4GH | https://www.ga4gh.org/news/webinar-recap-ga4gh-passports-benefits-of-integrating-a-global-electronic-id-for-accessing-biomedical-data-part-1/ |
| Webinar GA4GH Passports: Benefits of Integrating a Global Electronic ID for Accessing Biomedical Data (Part 1) Video | GA4GH | https://youtu.be/l5Cu76NQyUY |
| Webinar GA4GH Passports: Implementing GA4GH Passports and AAI: Technical Deep Dive (Part 2) Summary | GA4GH | https://www.ga4gh.org/news/webinar-recap-ga4gh-passports-implementing-ga4gh-passports-and-aai-technical-deep-dive-part-2/ |
| Webinar GA4GH Passports: Implementing GA4GH Passports and AAI: Technical Deep Dive (Part 2) Video | GA4GH | https://youtu.be/K7HID5KAhz0 |

### RESOURCE AVAILABILITY

**Lead contact**

Further information and requests for resources should be directed to and will be fulfilled by contacting Craig Voisin at craigv@google.com.

**Materials availability**

This work did not generate new unique reagents.

**Data and code availability**

All original code is publicly available at GitHub.

GA4GH Passport Standard v1.0: https://bit.ly/ga4gh-passport-v1

GA4GH AAI Profile Standard v1.0: https://bit.ly/ga4gh-aai-profile

GA4GH Passport Visa Format v1.0: https://bit.ly/ga4gh-visa-format-v1

MitreID ProxyIdP OIDC Server source code (https://doi.org/10.5281/zenodo.4877248)

■ Class responsible for constructing the claim from internal and external data: GA4GH Claim Source code (https://github.com/CESNET/perun-mitreid/blob/master/oidc-idp/src/main/java/cz/muni/ics/oidc/server/elixir/GA4GHClaimSource.java)

■ Utility class used for parsing the claim value and displaying it in a human-readable way: GA4GH Token Parser code (https://github.com/CESNET/perun-mitreid/blob/master/oidc-idp/src/main/java/cz/muni/ics/oidc/server/elixir/GA4GHTokenParser.java)

CESNET Perun Project source code (https://doi.org/10.5281/zenodo.5032084)

ELIXIR REMS source code (https://github.com/CSCfi/rems)

ELIXIR AAI GA4GH echo service source code (https://github.com/ICS-MU/ga4gh-passport-tester)

## METHOD DETAILS

The procedures listed are designed to determine if Passport Brokers, such as the ELIXIR Broker, are able to collect visas from systems like REMS and convey them to a Passport Clearinghouse, here represented as a Passport inspection service known as the "ELIXIR echo test service." These are publicly deployed infrastructure services and the procedure will check that general user accounts, whether at a supported research institution or more common public identity providers, may participate in the Passport visa collection and distribution process.

To prevent accidental use of existing authorization or account state maintained by the browser, you are advised to use an "incognito window" in the web browser when executing these steps.

1. Prerequisite: An account in an enabled research organization (e.g., Sanger Institute, Masaryk University) or the creation of a test gmail (Google), LinkedIn, or ORCID account.

2. Visit the ELIXIR registration service (https://elixir-europe.org/register).

   a. Use the sign-in service and account established in the previous step.
   b. You must agree to the acceptable use policy to proceed.
   c. Complete the registration process.
   d. You may ensure your account is active by visiting the ELIXIR profile page (https://profile.aai.elixir-czech.org).

3. Visit the ELIXIR sensitive data access management test system (https://sd-apply.csc.fi/%20 (beta)) instance of REMS.

   a. Login using your ELIXIR account from the previous step.
   b. Locate the "ELIXIR Beacon Network - Test Dataset" and click the "Add to Cart" button. It may help if you search for this dataset first.
   c. Click the "Apply" button near the top of the page. The Application page will appear with information about the applicant (you) and resource (ELIXIR Beacon Network - Test Dataset).
   i. Click the "Accept the terms of use" (assuming you agree to the terms in a test context).
   ii. In the "Description" field, add the text "Testing the Passport visa system within ELIXIR."
   iii. Under the "Actions" section, click the "Send application" button.
   iv. At the top of the page, verify that the "state" shows green arrows with checkmarks for all three parts: "apply," "approval," and "approved." You are ready for the next step when the "approved" label receives the green checkmark.

4. Visit the ELIXIR echo test service (https://echo.aai.elixir-czech.org/).

   a. Sign in using the ELIXIR account created in step #2 above.
   b. View the Passport summary contents - this is the "basic view."
   c. Click the "Switch to expert view" button.
   d. Observe any visas the Passport has in the tables provided.

e. In the "ControlledAccessGrants Visas" section, click the "Display raw decoded JWT" button for the visa shown. For example, it should look something like this with different timestamp numbers (seconds since unix epoch) in the "iat," "exp," and "asserted" fields:

```
{
"alg": "RS256,"
"jku": "https://sd-apply.csc.fi/api/jwk,"
"typ": "JWT,"
"kid": "<key-identifier>"
}
.
{
"iss": "https://sd-apply.csc.fi/,"
"sub": "<identity-number>@elixir-europe.org,"
"iat": 1625144975,
"exp": 1656680975,
"ga4gh_visa_v1": {
"type": "ControlledAccessGrants,"
"value": "https://beaconpy-elixirbeacon.rahtiapp.fi/urn:hg:example-controlled,"
"source": "https://sd-apply.csc.fi/,"
"by": "dac,"
"asserted": 1625055001
}
}
```

f. To verify that your Passport complies with the GA4GH Passport Standard v1.0 (https://bit.ly/ga4gh-passport-v1) and GA4GH AAI Profile Standard v1.0 (https://bit.ly/ga4gh-aai-profile), follow the links from the Key resources table. Note that conformance of the output from the previous sub-step is specified by the GA4GH Passport Visa Format v1.0 (https://bit.ly/ga4gh-visa-format-v1) section. Fields may appear in any order within their designated block.

Implementations like ELIXIR and its related partners have:

1. Implemented visa generation in compliance with the GA4GH Passport standard.
2. Are able to use identities from a wide variety of research and general public infrastructure providers that already exist.
3. Are able to encode the visas into Passports and deliver them between systems while leveraging existing standards, such as OIDC that is used during this procedure.
4. Visas that are delivered to a Passport Clearinghouse retain digitally verifiable permissions from the original source of authority (i.e., the DAC approver in this case).

