## [Document S2. Transparent peer review records for Voisin et al · Cell Genomics]

GA4GH Passport standard for digital identity and access permissions

Craig Voisin¹ *, Mikael Linden^{2,3} *, Stephanie O.M. Dyke⁴ *, Sarion R. Bowers⁵ *, Kathy Reinold⁶, Jonathan Lawson⁶, Stephanie Li^{6,7}, Vivian Ota Wang⁸, Maxmillian P. Barkley⁹, David Bernick⁶, Mélanie Courtot¹⁰, Francis Jeanson¹¹, Jianpeng Chao¹, Melissa A. Konopko^{5,7}, Martin Kuba¹², Jaakko Leinonen², Anthony A. Philippakis⁶, Pinar Alper¹³, Gregory A. Rushton⁶, J. Dylan Spalding^{2,3}, Juha Törnroos^{2,3}, Ilya Tulchinsky¹⁴ *, Jaime M. Guidry Auvil⁸, Tommi H. Nyrönen^{2,3}

Summary

Scientific Editor:	Orli Bahcall
Initial submission:	3/18/2021
Revision received:	7/08/2021
Accepted:	9/02/2021
Rounds of review:	2
Number of reviewers:	4

Referee reports, first round of review

Reviewer #1: The paper successfully explains how the GA4GH Passport technology builds on well established AAI layers to enable secure data access which creates clear added-value for the field. Some readers may question the use of the term 'standard approved in 2019' for GA4GH since it could give the impression that it has been approved by an international standards body, such as ISO, that is recognised by national governments. While the GA4GH has very reputable members, it appears to be a community based alliance.

The ELIXIR example implementation is cited but it is not clear if the RAS, GEM & H3Africa implementations exist and can inter-operate. If this were the case it would increase the impact and provide a stronger justification for being considered a recognised standard.

The GA4GH Passport could have value beyond genomics and health.

Reviewer #2: I am reviewing this paper not as an expert in Human Genomics but as a researcher with expertise in Data Science, Trust, Security and Identity Management.

This paper, "GA4GH Passport data access technology standard for distributed genomics & health research", is the presentation of an important new technical data science standard aimed at enabling the secure sharing of human genomic data between trusted researchers. The GA4GH Passport together with the use of the associated federated Authentication and Authorisation (AAI) techniques is a very important step in achieving

the stated

aims of the Global Alliance for Genomics & Health: "Enabling responsible genomic data sharing for the benefit of human health". The standard developed builds on the earlier work defining "registered access" to the genomics data. The authors, in addition to describing the methods and components of the Passport standard, also present an example of an implementation of the Passport Broker by ELIXIR to retrieve ControlledAccessGrants visas from European Genome-phenome Archive, together with the ELIXIR AAI, integrated with the eduGAIN inter-federation, enabling researchers to "login" using their home University identity credentials. All of this is a very important automation of the trust relationships involved in access to and sharing of genomic data for the benefit of human health.

The GA4GH Passport standard is a very important contribution to Data Science in general, not only for the benefit of the Genomics researchers and clinicians, but also as an exemplar of great interest for other scientific disciplines that need a similar solution to their own problems of enabling easy and trusted access to and sharing of their own sensitive data.

I enjoyed reading both the paper itself and the approved GA4GH Passport version 1 standard referenced by the authors as reference [12] and mentioned directly in their submission. I also studied the related GA4GH AAI OpenID Connect Profile. I am not an expert in the deployment and use of OpenID Connect technologies or use of JSON Web Tokens so I am not able to review the detailed architecture of the profile and protocols. Everything I have read, however, makes very good sense and I see no technical problems or security concerns that I should highlight. My comments and suggestions here below relate just to the text of the submitted paper and not to the approved standard itself.

I have no hesitation in recommending this paper, with its clear and well written presentation and the nice and easy to understand figures, for publication in Cell Genomics. It could be published "as is", but I do have few suggestions of minor additions or changes that could be made by the authors to improve this presentation of the Passport Standard and its use.

General suggestions:

1. I was expecting to see some description of how Trust is established between the various components of the Passport workflow. How and why should the Clearinghouse trust the Broker? How does the Broker establish a list of trusted Visa issuers? Are there trust issues in the AAI with the University Identity Provider? Does the Researcher's Authentication have to meet some Level of Assurance requirements to be sufficiently trust-worthy? I am sure these issues have all been considered during the design of the whole approach. I suggest that a paragraph or two addressing some of these operational trust issues would enhance the paper.
Or the authors should at least state that these trust issues are a matter to be addressed later by the entities deploying and using the components. In that latter case some guidance of how to solve the trust issues would be useful.
2. You state that a Visa can be revoked. How is this done? Or how is revocation supported in the protocol? Or is this again a matter to be decided at time of deployment?

More minor specific comments:

3. The references are not referred to in numerical order through the paper.
4. In the section on "Tiers of Access" - Open Access: You refer to Creative Commons Zero as an example, but just refer to it as CC-0. Perhaps it is better to spell it out in full and/or give a reference to Creative Commons Zero.
5. In the section titled "GA4GH Passport Standard" end of the first paragraph after [Figure 4] the words "data access to as intended" is not clear. Either the word "to" needs to go or the missing word following "access to" needs inserting.
6. In the Author Contributions and Funding sections I see at least one example where author initials "JL" are used. There are two authors with these initials. I have not checked all the other initials used.

Reviewer #3: This article provides a very good overview of the new GA4GH Passport data access technology standard for distributed genomics and health research. The article organization and writing are very good, and the 7 figures are all very helpful in understanding the topic. This new technology standard has significant potential to ease genomic data sharing while maintaining appropriate security controls.

As a Research Article, the experimental evaluation and comparison with related work is lacking. This topic may be more appropriate as a Technology Article, but in that case, the demonstration of "significant improvements to existing methods" is lacking. For example, the article indicates that NIH dbGaP has adopted the new technology standard, but no comparison is made with the existing NIH dbGaP Authorized Access System (<https://dbgap.ncbi.nlm.nih.gov/aa>). Comparison with federated access to genomics datasets via Galaxy (<https://doi.org/10.1093/bioinformatics/btz472>) would also be enlightening. A discussion of potential impact beyond the genomics community, such as support for access to proprietary astronomical data (<https://ls.st/RDO-013>) would strengthen the article to match the journal's requirement for Research Articles to "report conceptual advances or discoveries that will be of unusual significance to progressing research in the genomics community and beyond."

Regarding the "Code and Algorithms" contribution, the https://github.com/ga4gh-duri/ga4gh-duri.github.io/blob/master/researcher_ids/ga4gh_passport_v1.md site provides a technical specification but not computer code. Is there a reference implementation that could be cited and evaluated for reproducibility, perhaps from ELIXIR? Also, the <https://echo.aai.elixir-czech.org/> site provided under the "Code and Algorithms" contribution requires a login, so it does not meet the journal submission requirement which states that "reviewers must have free access to your custom code and new algorithms without compromising their anonymity."

The abstract states that "The GA4GH Passport is currently implemented on research infrastructures (e.g. ELIXIR Europe) and commercial services (e.g. Google)." However, the text only describes the ELIXIR implementation. Also, the text states that "the European Genome-Phenome Archive (EGA) [1] and the NIH Database of Genotypes and Phenotypes (dbGaP) [2, 3], are using GA4GH Passport standards in privacy preserving data infrastructures that communicate access rights for the data they hold." However, only the ELIXIR EGA implementation is described in detail. Adding detailed descriptions of the Google and NIH dbGaP implementations would strengthen the article by demonstrating multiple (ideally independent and interoperable) implementations of the technical standard.

The article describes in depth how GA4GH Passports can contain visas issued by multiple sources of authority but does not sufficiently describe how the Passport Clearinghouse verifies the visas. How do data stewards decide which sources of authority (issuers) to trust? How does the Passport Clearinghouse know that a visa was issued by the proper authority? Is there a federation or trust framework that binds these multiple sources of authority together? The article mentions that "a trust framework between different DACs could be developed" as future work, but that seems to address only one aspect of the trust that is required between visa issuers and consumers (data providers).

By my count, the article is currently under 25,000 characters, which matches the length requirement for a Short Article. Thus, I think the authors have space to expand to a full Article to address the above topics.

Reviewer #4: The manuscript addresses the important challenge of communicating identities and permissions for data access across a federated system. The introduced GA4GH Passport describes a standard adopted by the Global Alliance for Genomics and Health GA4GH that allows to provide information on identity and approved access to datasets across institutions and functions in a federated network.

It is stated that public trust is a crucial requirement and claim that the Passport increases trust but provide no further explanation how such trust can be derived from the passport. In particular, no discussion on the security and potential vulnerabilities takes place. For security, there is merely a reference to security and privacy best practices in the context of GA4GH AAI specifications but neither references are given nor how such implementation will take place and if e.g. vulnerabilities could be created as this implementation is left to the networks when setting up their authorisation framework. As trust is a major issue for secondary use of genomic data, a section on the security framework, potential vulnerabilities and the role of and dependency on the the entities implementing the passport should therefore be added.

The classical data access procedures are very well described. However, a state of the art gap analysis of challenges to overcome could be added to motivate better the pressure that led to the development of the Passport. However, this gap analysis should complete and not limited to those elements solved by the Passport. Subsequently, this can be taken up again in the Discussion Section and the Passport matched on how well it serves the needs of the community for secondary use of data.

Another element missing in the manuscript is a description of the access procedures based on the GA4GH Passport, including the information of when a Passport is acquired, who issues the passport etc. The procedure is sketched in figure 3 but detailed information on who, what and by whom in the entire chain of events is insufficient. In particular, there seem to be discrepancies between fig. 3 and fig. 5. In fig. 3, the researcher has a passport already at the time of the first login when browsing the database before requesting access while in fig. 5, the passport is only acquired by the researcher when access to the data is established. This can be explained if the endpoint of fig. 5 (access) is equivalent to the first step in fig. 3 (browsing the database) but the choice of wording (browsing versus access) does not support this in an intuitive way. Similarly, there is also confusion in the wording between fig. 3 and fig. 2 as in fig. 3 data discovery is mentioned as a step after DAC approval while in fig. 2, data discovery is the step before applying for access. Intuitive interpretation is further hampered by the fact that identities are visas on the passport where visas normally code permissions only.

These inconsistencies and the lack of a detailed step by step example of the procedures, workflows and example content stored on the Passport through the different visas make it very difficult for readers not familiar with the Passport to understand the mode of operation. Providing concrete examples of visas matching the examples would further increase the value of the example for the reader. It is therefore strongly recommended that such detailed step by step example is provided with hypothetical entities and including a realistic potential content of the visas and how such scenario would be implemented in the coding of the visas. It is also recommended to define some key terms used in the text in a separate definition box.

More information and a solid discussion of the different features of the Passport would further improve the manuscript. It is stated that the Passport provides a high degree of flexibility to meet different policy needs. Here, it should be discussed in the text how this is achieved and what the price is for which this flexibility is achieved. Does this mean an organisation will need different Passport Clearinghouses for an entity participating in different networks as Passports may not be established based on the same definitions? Where does information in the visas build on standards, where on proprietary definitions? Is all information in visas to be machine readable or is free text included? And in consequence, does the flexibility come at the expense of less machine readability of policy options?

The actual section "Discussion" is focussed on existing and potential implementations of the GA4GH Passport and potential future developments. The conclusions are thus not derived from any discussion in this section but seem to be standalone statements that are not sufficiently motivated. A strong recommendation is that the current discussion section is split into a part on an outlook section covering implementations and future development, while an actual discussion is added that picks up all the statements of the current conclusion and motivates them - how well do Passports serve the actual needs (see also the recommendation above to include a state of the art and requirement section)? Which needs are not yet or only partially served? What are the strengths of the Passport? What the weaknesses? Where are trade-offs being made? What are the consequences of these trade-offs? Could the dependency on the entire GA4GH framework hamper the implementation for some organisations? Are there competing approaches?

On a side note: I would like to strongly discourage the use of the term data owner. For personal data, there are no data owners but merely entities with rights in data. The term data provider or data custodian may be more appropriate.

Author response to the first round of review

Reviewer #1: The paper successfully explains how the GA4GH Passport technology builds on well established AAI layers to enable secure data access which creates clear added-value for the field. Some readers may question the use of the term 'standard approved in 2019' for GA4GH since it could give the impression that it has been approved by an international standards body, such as ISO, that is recognised by national governments. While the GA4GH has very reputable members, it appears to be a community based alliance.

The ELIXIR example implementation is cited but it is not clear if the RAS, GEM & H3Africa implementations exist and can inter-operate. If this were the case it would increase the impact and provide a stronger justification for being considered a recognised standard.

The GA4GH Passport could have value beyond genomics and health.

Response:

- Edited text about NIH/RAS, ASI.
- Included more references.
- Linden M et al. "Common ELIXIR Service for Researcher Authentication and Authorisation" F1000Research. <https://f1000research.com/articles/7-1199>
- Github GA4GH AAI <https://github.com/ga4gh/data-security/blob/master/AAI/AAIConnectProfile.md>

Reviewer #2: I am reviewing this paper not as an expert in Human Genomics but as a researcher with expertise in Data Science, Trust, Security and Identity Management.

This paper, "GA4GH Passport data access technology standard for distributed genomics & health research", is the presentation of an important new technical data science standard aimed at enabling the secure sharing of human genomic data between trusted researchers. The GA4GH Passport together with the use of the associated federated Authentication and Authorisation (AAI) techniques is a very important step in achieving the stated aims of the Global Alliance for Genomics & Health: "Enabling responsible genomic data sharing for the benefit of human health". The standard developed builds on the earlier work defining "registered access" to the genomics data. The authors, in addition to describing the methods and components of the Passport standard, also present an example of an implementation of the Passport Broker by ELIXIR to retrieve ControlledAccessGrants visas from European Genome-phenome Archive, together with the ELIXIR AAI, integrated with the eduGAIN inter-federation, enabling researchers to "login" using their home University identity credentials. All of this is a very important automation of the trust relationships involved in access to and sharing of genomic data for the benefit of human health.

The GA4GH Passport standard is a very important contribution to Data Science in general, not only for the benefit of the Genomics researchers and clinicians, but also as an exemplar of great interest for other scientific disciplines that need a similar solution to their own problems of enabling easy and trusted access to and sharing of their own sensitive data.

I enjoyed reading both the paper itself and the approved GA4GH Passport version 1 standard referenced by the authors as reference [12] and mentioned directly in their submission. I also studied the related GA4GH AAI OpenID Connect Profile. I am not an expert in the deployment and use of OpenID Connect technologies or use of JSON Web Tokens so I am not able to review the detailed architecture of the profile and protocols. Everything I have read, however, makes very good sense and I see no technical problems or security concerns that I should highlight. My comments and suggestions here below relate just to the text of the submitted paper and not to the approved standard itself.

I have no hesitation in recommending this paper, with its clear and well written presentation and the nice and easy to understand figures, for publication in Cell Genomics. It could be published "as is", but I do have few suggestions of minor additions or changes that could be made by the authors to improve this presentation of the Passport Standard and its use.

General suggestions:

1. I was expecting to see some description of how Trust is established between the various components of the Passport workflow. How and why should the Clearinghouse trust the Broker? How does the Broker establish a list of trusted Visa issuers? Are there trust issues in the AAI with the University Identity Provider? Does the Researcher's Authentication have to meet some Level of Assurance requirements to be sufficiently trust-worthy? I am sure these issues have all been considered during the design of the whole approach. I suggest that a paragraph or two addressing some of these operational trust issues would enhance the paper.

Or the authors should at least state that these trust issues are a matter to be addressed later by the entities deploying and using the components. In that latter case some guidance of how to solve the trust issues would be useful.

2. You state that a Visa can be revoked. How is this done? Or how is revocation supported in the protocol? Or is this again a matter to be decided at time of deployment?

More minor specific comments:

3. The references are not referred to in numerical order through the paper.

4. In the section on "Tiers of Access" - Open Access: You refer to Creative Commons Zero as an example, but just refer to it as CC-0. Perhaps it is better to spell it out in full and/or give a reference to Creative Commons Zero.

5. In the section titled "GA4GH Passport Standard" end of the first paragraph after [Figure 4] the words "data access to as intended" is not clear. Either the word "to" needs to go or the missing word following "access to" needs inserting.

6. In the Author Contributions and Funding sections I see at least one example where author initials "JL" are used. There are two authors with these initials. I have not checked all the other initials used.

Response: Improved multiple parts of the manuscript.

- Included extensive STAR and SI chapters.
- Provided references to implementations in the text.
- A public version and source of the Passport standard is in Github.
- Open public implementation is usable via ELIXIR.

Reviewer #3: This article provides a very good overview of the new GA4GH Passport data access technology standard for distributed genomics and health research. The article organization and writing are very good, and the 7 figures are all very helpful in understanding the topic. This new technology standard has significant potential to ease genomic data sharing while maintaining appropriate security controls.

As a Research Article, the experimental evaluation and comparison with related work is lacking. This topic may be more appropriate as a Technology Article, but in that case, the demonstration of "significant improvements to existing methods" is lacking. For example, the article indicates that NIH dbGaP has adopted the new technology standard, but no comparison is made with the existing NIH dbGaP Authorized Access System (<https://dbgap.ncbi.nlm.nih.gov/aa>). Comparison with federated access to genomics datasets via Galaxy (<https://doi.org/10.1093/bioinformatics/btz472>) would also be enlightening. A discussion of potential impact beyond the genomics community, such as support for access to proprietary astronomical data (<https://ls.st/RDO-013>) would strengthen the article to match the journal's requirement for Research Articles to "report conceptual advances or discoveries that will be of unusual significance to progressing research in the genomics community and beyond."

Regarding the "Code and Algorithms" contribution, the https://github.com/ga4gh-duri/ga4gh-duri.github.io/blob/master/researcher_ids/ga4gh_passport_v1.md site provides a technical specification but not computer code. Is there a reference implementation that could be cited and evaluated for reproducibility, perhaps from ELIXIR? Also, the <https://echo.aai.elixir-czech.org/> site provided under the "Code and Algorithms" contribution requires a login, so it does not meet the journal submission requirement which states that "reviewers must have free access to your custom code and new algorithms without compromising their anonymity."

The abstract states that "The GA4GH Passport is currently implemented on research infrastructures (e.g. ELIXIR Europe) and commercial services (e.g. Google)." However, the text only describes the ELIXIR implementation. Also, the text states that "the European Genome-Phenome Archive (EGA) [1] and the NIH Database of Genotypes and Phenotypes (dbGaP) [2, 3], are using GA4GH Passport standards in privacy preserving data infrastructures that communicate access rights for the data they hold." However, only the ELIXIR EGA implementation is described in detail. Adding detailed descriptions of the Google and NIH dbGaP implementations would strengthen the article by demonstrating multiple (ideally independent and interoperable) implementations of the technical standard.

The article describes in depth how GA4GH Passports can contain visas issued by multiple sources of authority but does not sufficiently describe how the Passport Clearinghouse verifies the visas. How do data stewards decide which sources of authority (issuers) to trust? How does the Passport Clearinghouse know that a visa was issued by the proper authority? Is there a federation or trust framework that binds these multiple sources of authority together? The article mentions that "a trust framework between different DACs could be developed" as future work, but that seems to address only one aspect of the trust that is required between visa issuers and consumers (data providers).

By my count, the article is currently under 25,000 characters, which matches the length requirement for a Short Article. Thus, I think the authors have space to expand to a full Article to address the above topics.

Response:

- Abstract and Introduction revised.
- Figure 3 updated to better illustrate Passport in the GA4GH context.
- Added ASI and NIH use cases leveraging Passport concept for relevant and currently operating data access processes.
- Added reference to Galaxy (software) service operations aiming to leverage federated AAI solution for data access.
- ELIXIR AAI's code repositories and description:
<https://docs.google.com/document/d/1p3Fqd50jFg10btrrB76fNQTMkeLXANIZ1tneIsQCus/edit?usp=sharing>

Reviewer #4: The manuscript addresses the important challenge of communicating identities and permissions for data access across a federated system. The introduced GA4GH Passport describes a standard adopted by the Global Alliance for Genomics and Health GA4GH that allows to provide information on identity and approved access to datasets across institutions and functions in a federated network.

It is stated that public trust is a crucial requirement and claim that the Passport increases trust but provide no further explanation how such trust can be derived from the passport. In particular, no discussion on the security and potential vulnerabilities takes place. For security, there is merely a reference to security and privacy best practices in the context of GA4GH AAI specifications but neither references are given nor how such implementation will take place and if e.g. vulnerabilities could be created as this implementation is left to the networks when setting up their authorisation framework. As trust is a major issue for secondary use of genomic data, a section on the security framework, potential vulnerabilities and the role of and dependency on the the entities implementing the passport should therefore be added.

The classical data access procedures are very well described. However, a state of the art gap analysis of challenges to overcome could be added to motivate better the pressure that led to the development of the Passport. However, this gap analysis should complete and not limited to those elements solved by the Passport. Subsequently, this can be taken up again in the Discussion Section and the Passport matched on how well it serves the needs of the community for secondary use of data.

Another element missing in the manuscript is a description of the access procedures based on the GA4GH Passport, including the information of when a Passport is acquired, who issues the passport etc. The procedure is sketched in figure 3 but detailed information on who, what and by whom in the entire chain of events is insufficient. In particular, there seem to be discrepancies between fig. 3 and fig. 5. In fig. 3, the researcher has a passport already at the time of the first login when browsing the database before requesting access while in fig. 5, the passport is only acquired by the researcher when access to the data is established. This can be explained if the endpoint of fig. 5 (access) is equivalent to the first step in fig. 3 (browsing the database) but the choice of wording (browsing versus access) does not support this in an intuitive way. Similarly, there is also confusion in the wording between fig. 3 and fig. 2 as in fig. 3 data discovery is mentioned as a step after DAC approval while in fig. 2, data discovery is the step before applying for access. Intuitive interpretation is further hampered by the fact that identities are visas on the passport where visas normally code permissions only.

These inconsistencies and the lack of a detailed step by step example of the procedures, workflows and example content stored on the Passport through the different visas make it very difficult for readers not

familiar with the Passport to understand the mode of operation. Providing concrete examples of visas matching the examples would further increase the value of the example for the reader. It is therefore strongly recommended that such detailed step by step example is provided with hypothetical entities and including a realistic potential content of the visas and how such scenario would be implemented in the coding of the visas. It is also recommended to define some key terms used in the text in a separate definition box.

More information and a solid discussion of the different features of the Passport would further improve the manuscript. It is stated that the Passport provides a high degree of flexibility to meet different policy needs. Here, it should be discussed in the text how this is achieved and what the price is for which this flexibility is achieved. Does this mean an organisation will need different Passport Clearinghouses for an entity participating in different networks as Passports may not be established based on the same definitions? Where does information in the visas build on standards, where on proprietary definitions? Is all information in visas to be machine readable or is free text included? And in consequence, does the flexibility come at the expense of less machine readability of policy options?

The actual section "Discussion" is focussed on existing and potential implementations of the GA4GH Passport and potential future developments. The conclusions are thus not derived from any discussion in this section but seem to be standalone statements that are not sufficiently motivated. A strong recommendation is that the current discussion section is split into a part on an outlook section covering implementations and future development, while an actual discussion is added that picks up all the statements of the current conclusion and motivates them - how well do Passports serve the actual needs (see also the recommendation above to include a state of the art and requirement section)? Which needs are not yet or only partially served? What are the strengths of the Passport? What the weaknesses? Where are trade-offs being made? What are the consequences of these trade-offs? Could the dependency on the entire GA4GH framework hamper the implementation for some organisations? Are there competing approaches?

On a side note: I would like to strongly discourage the use of the term data owner. For personal data, there are no data owners but merely entities with rights in data. The term data provider or data custodian may be more appropriate.

Response:

- Abstract and Introduction revised.
- Figure 3 updated to better illustrate Passport in the GA4GH context.
- Included extensive STAR and SI chapters.
- Provided references to implementations in the text.
- A public version and source of the Passport standard is in Github.
- Open public implementation is usable via ELIXIR.
- Improved term descriptions.
- Introduced a new table describing organizational roles challenges and benefits.
- Introduced more examples in "results" describing benefits in action.
- Improved figures showing more technical systems workflow.
- STAR METHODS walking through user journey.

Referee reports, second round of review

Reviewer #1: The modifications since the first review have significantly improved the readability and clarity of the paper.

Reviewer #2: I have reviewed this revision of the paper not as an expert in Human Genomics but as a researcher with expertise in Data Science, Trust, Security and Identity Management.

This revision of the paper has successfully addressed all of the concerns and issues given in my earlier review. The new layout of the paper, the additional detail and the updated figures all contribute to an excellent presentation of the design, the technical details, the issues of trust and the potential impact of the GA4GH Passport technology. I fully agree with the Conclusions of the paper and I still feel strongly that the GA4GH Passport standard is a very important contribution to Data Science in general, not only for the benefit of the Genomics researchers and clinicians, but also as an exemplar of great interest for other disciplines.

I successfully followed all steps in the STAR Method. This was easy to execute and gave good insight into the technical detail of the passports, visas and associated processes.

I conclude that this paper is now fully ready for publication and I have not identified any additional required changes.

Reviewer #3: This article introduces the new GA4GH Passport data access technology standard for distributed genomics and health research. The article organization and writing are very good, and the 7 figures and 3 tables are all very helpful in understanding the topic. This new technology standard has significant potential to ease genomic data sharing while maintaining appropriate security controls. The article describes 3 implementations (ELIXIR AGE, NIH dbGaP, and ASI MSSNG) which demonstrate significant adoption and effective use of the standard.

This revised manuscript addresses all issues raised in my review of the original submission.

Reviewer #4: The manuscript is considerably improved after the revisions and will be appealing to a much broader audience. I am sure that it will become a guidance for many in the field.

Author response to the second round of review

1) Please respond to the referee comments and incorporate requested revisions.

Done.

2) Please see the attached file, which includes my detailed edits to the main text, all noted with tracked changes and/or Comments. These edits are intended to improve the clarity, presentation and reporting in the manuscript.

- Please note that I have queried often to clarify the meaning and to introduce new terms or concepts or related work. Please keep in mind that we are editing in order to make this work more accessible to a

general reader, so that the work will more broadly read and used. You may find it helpful to have other colleagues comment and suggest further revisions to improve the presentation and accessibility, and to coordinate with the other GA4GH manuscripts to be published in this special issue.

Response: We have carefully responded to all editorial comments, and agree they have made the presentation better.

- I have also requested more detailed description of the methods; this should be included in relevant sections of the text, STAR Methods, and in additional Supplementary. Please include complete documentation for all of the work reported in the current manuscript and for all components of **Passport**. There needs to be complete and clear documentation to the extent that a general reader can understand and repeat all of this work to the same standards, supporting transparency and reproducibility.

Response: Structure of the narrative was improved to explain better how the requirements for federated data sharing lead to the Passport standard, how the Passport service components were designed and how they are implemented. Figures (3, 4 and 7) were upgrade to respond to editorial comments to give technical details of the message exchange “protocols”. GA4GH Github provides further details.

- For all display items: Please include a full explanatory legend title and text, to explain the full content of the figure, so that the content, context and messages are easily understood by the general reader without needing to read the main text.

Response: Done.

2) Please format according to our Technology article format

Response: Done

3) Please include a cover letter that details all changes made in the revised manuscript (you may also include an additional manuscript file with tracked changes).

Response: Requested edits triggered multiple rounds of revision by the authors. Unfortunately Word track changes did not have the capacity to track this kind of process. The only feasible response we can do to provide on clean and rev.1 docs with comments and the the online googledoc where the manuscript history can be reviewed if necessary.

4) Limitations of the Study: Thank you for including this section, I have provided some suggestions for how to edit and what to include in this section. Our general guidance for use of this section is:

- Please include a paragraph or two as a subsection in the Discussion entitled "Limitations of the Study", highlighting potential caveats of the work. The goal of this section is to promote clarity and transparency by highlighting any limitations in the interpretation of the study, including limits of the techniques used and/or assumptions made. It can include additional experiments that would be necessary to definitively prove some conclusions but should be specific to the paper. Examples include sample size, genetic strains, detection levels, etc. The “Limitations of this Study” paragraph should be specific to this paper.

Response: Chapter revised.

5) Please include a 'Highlights and eToc' and 'Graphical Abstract' with your revision. These should describe the context and significance of the work for a broad readership. The goal is to highlight the major advances in the paper in order to attract the attention of the non-specialist, without including extensive detail.

Response: Revised according to journal format.

Reviewers' Comments:

Reviewer #1: The modification since the first review have significantly improved the readability and clarity of the paper.

Response: Acknowledged. Thank you.

Reviewer #2: I have reviewed this revision of the paper not as an expert in Human Genomics but as a researcher with expertise in Data Science, Trust, Security and Identity Management.

This revision of the paper has successfully addressed all of the concerns and issues given in my earlier review. The new layout of the paper, the additional detail and the updated figures all contribute to an excellent presentation of the design, the technical details, the issues of trust and the potential impact of the GA4GH **Passport** technology. I fully agree with the Conclusions of the paper and I still feel strongly that the GA4GH **Passport** standard is a very important contribution to Data Science in general, not only for the benefit of the Genomics researchers and clinicians, but also as an exemplar of great interest for other disciplines.

I successfully followed all steps in the STAR Method. This was easy to execute and gave good insight into the technical detail of the passports, visas and associated processes.

I conclude that this paper is now fully ready for publication and I have not identified any additional required changes.

Response: Acknowledged. Thank you.

Reviewer #3: This article introduces the new GA4GH **Passport** data access technology standard for distributed genomics and health research. The article organization and writing are very good, and the 7 figures and 3 tables are all very helpful in understanding the topic. This new technology standard has significant potential to ease genomic data sharing while maintaining appropriate security controls. The article describes 3 implementations (ELIXIR AGE, NIH dbGaP, and ASI MSSNG) which demonstrate significant adoption and effective use of the standard.

This revised manuscript addresses all issues raised in my review of the original submission.

Response: Acknowledged. Thank you.

Reviewer #4: The manuscript is considerably improved after the revisions and will be appealing to a much broader audience. I am sure that it will become a guidance for many in the field.

Response: Acknowledged. Thank you.